# Trajectories of Emotional Exhaustion and Their Contribution to Depression: Optimism as a Buffer in Young People

**DOI:** 10.3390/brainsci15060656

**Published:** 2025-06-18

**Authors:** Martha Cruz-Soto, Emmanuel Said Baeza-Torres, Luis Castañeda Pelaez, Jesús Rojas Jaimes, Jorge Palacios-Delgado

**Affiliations:** 1Coordinación Nacional de Investigación en Ciencias de la Salud, Universidad del Valle de México, Av. Marina Nacional 180, Anáhuac I Secc, Miguel Hidalgo, Ciudad de México 11320, Mexico; martha.cruzso@uvmnet.edu; 2Facultad de Psicología, Universidad Tecnológica de México, Av. Marina Nacional 162, Anáhuac I Secc, Miguel Hidalgo, Ciudad de México 11320, Mexico; emmanuel_baeza@my.uvm.edu.mx; 3Facultad de Ciencias de la Salud, Universidad Privada del Norte, Lima 15314, Peru; luisneda_69@hotmail.com (L.C.P.); jesus.rojas.jaimes@gmail.com (J.R.J.); 4Unidad de Investigación en Neurociencias, Facultad de Psicología, Campus Querétaro, Universidad del Valle de México, Querétaro 76230, Mexico

**Keywords:** depressive symptoms, optimism, emotional exhaustion, path model

## Abstract

Background: In Mexico, depression is one of the main mental health problems, with university students being particularly susceptible. Recent studies have explored the relationship between emotional factors and depression in young people. Our study investigates whether optimism buffers the indirect relationship between burnout, stress and coping in mitigating the negative effects on depressive symptoms in young university students. We hypothesized that optimism would moderate the negative impacts of stress and emotional exhaustion on depression. Methods: In total, 497 students of a university in Mexico participated (63% female and 36.6% male), ranging in age from 18 to 29. Students completed screenings for depression, emotional scales and optimism measures. Results: Emotional exhaustion and stress are direct predictors of depression. Although coping strategies did not have a direct effect, optimism mediated the relationship between stress and depression. Conclusions: These findings suggest that promoting optimism in university students could be an effective strategy to reduce depressive symptoms, especially in the context of socioemotional vulnerability.

## 1. Introduction

### 1.1. Stress

While stress has been studied for more than thirty years [1], it has only recently been understood [1] that prevailing stressors are related to psychosocial factors that distinctly impact different age groups. According to the World Health Organization, stress is the response of an individual to situations of physical, mental or emotional pressure that impacts the body as both negative effects and physiological changes. However, it is the way we react to stress that marks the impact on our mental health, for example, through maladaptive coping strategies such as detachment, avoidance and emotional suppression [2,3]. Thus, when the individual cannot resolve, adapt or adequately cope with situations that are perceived as stressful, this can generate chronic stress, the activation of which is maintained for an average period of six months even when there is no real danger or the stressor is no longer present. This can then trigger mental health problems such as anxiety or depressive spectrum disorders [4,5].

Stress is part of daily life and is a neurophysiological response to situations that challenge homeostasis. Different studies [1,6,7,8] have examined the effects of stress on emotion and cognition [7,9]. Chronic stress is associated with burnout, resulting from workplace or school overload in which coping strategies have failed, that can be characterized according to levels of emotional exhaustion [8,10]. 

### 1.2. Emotional Exhaustion 

University students may present chronic stress with poor outcomes in terms of performance and grades leading to desertion [9,11] and the risk of developing stress-related diseases. 

There is a high prevalence of emotional exhaustion among students as they progress in their curricula. In a sample of 868 medical students, the average level of emotional exhaustion was measured as 47.8 ± 11.0% [10,12,13,14]. Moreover, students deal with due dates for assignments and exams while also experiencing lifestyle changes, peer pressure and financial struggles [11,15]. Young adults have an increased risk of anxiety and depression, and this is especially concerning at the beginning of their professional career, as they may find themselves trapped in a cycle where mental health issues and work-related stress exacerbate one another [13,16].

Recent studies have explored the relationship between emotional factors and depression in young people. The COVID-19 pandemic has significantly impacted youth mental health, leading to increased stress, anxiety and depression [14,17]. Stressful life events are related to depressive symptoms in adolescents, with higher stress levels in females associated with more severe depressive symptoms [15,18]. Research has found a positive correlation between emotional exhaustion and depression [16,19], where emotional exhaustion—the core component of burnout—is positively associated with depressive symptoms [17,20]. Finally, people with burnout and depressive episodes report severe levels of depressive symptoms [2,18].

### 1.3. Stress Management

Studies have found that anxiety, with poor coping strategies, makes young adults more prone to high levels of stress. Some students adopt coping strategies that rely on avoidance, substance abuse or denial, which leads to them experiencing higher levels of stress compared to those who adopt positive coping strategies, such as asking for social or professional support or participating in physical activities [4,19]. 

The effectiveness of the stress coping strategy will result in positive or negative outcomes for the person experiencing stress, and managing emotions may then enhance problem-solving abilities, whereas anger, frustration or denying reality can be harmful; these, respectively, represent active and passive strategies of managing stress. Expressing and identifying emotions may also contribute to better outcomes and may reduce depression-like symptoms [20,21].

### 1.4. Depressive Symptomatology

As stated in DSM-5, depressive symptoms include feeling sad, having lost interest in daily activities, problems sleeping, feeling tired, decreasing ability to concentrate and changes in appetite. Within the contextual model, depression can be generated and maintained by a decrease in behaviors that are reinforcing for the individual and an increase in avoidant coping behaviors that are maintained both by negative reinforcement (e.g., stopping activities in the face of unpleasant emotions) and positive reinforcement (e.g., immediate positive contingencies despite the negative consequences associated with depression). If, at this point, the individual acts in an avoidant manner based on their emotional state, then a spiral of depressogenic behaviors will be initiated in them that will intensify their depressive emotional state [21,22].

### 1.5. Moderation Effect of Optimism

Optimism is an emotional repertoire of personal resources that makes it easy to visualize, respond to and buffer life’s circumstances in a positive way [3]. Optimism has been shown to have a significant impact on mitigating the symptoms of depression [5]. Studies on positive psychology have revealed that optimists tend to experience fewer symptoms associated with depression [23,24]. Furthermore, optimism encourages coping, allowing individuals to more effectively overcome adversity [25,26]. The background research has shown that optimism buffers the effect of stress and depressive symptoms [27,28,29]; that is, when a person experiences emotional exhaustion, optimism works as a psychological resource that helps people cope by utilizing positive emotional states, which decreases the risk of depressive symptoms manifesting [3,5].

The studies indicate that optimism is an intrapersonal coping tool that can offset the adverse effects of stressors [5,22,30,31]. For example, optimism is associated with better psychological resistance to cope with stress, in addition to influencing the way in which individuals interpret and process negative experiences [30,32]. Individuals with higher levels of optimism are reportedly also more likely to engage in preventive health behaviors [31,32,33,34] in addition to having a lower risk of chronic diseases [33,34,35,36] and coping more effectively with stress and anxiety. An increase in depression rates among university students has been documented in previous studies. Liselotte and collaborators compared the prevalence of burnout and depression among medical students from a group of 4402 medical students and found that 58.2% screened positive for depression [36,37,38], highlighting the need to investigate the underlying mechanisms of these problems and to identify protective factors or strategies [39,40]. Understanding these factors is essential for designing effective interventions to promote emotional well-being [35,39].

The purpose of this study is to determine the effect of emotional exhaustion, stress and stress management on depressive symptoms as well as investigate whether optimism buffers depressive symptoms. Based on earlier research, optimism provides people with affective and behavioral resources that protect against the risk of manifesting depressive symptoms. Therefore, we investigated whether optimism buffers these relationships in mitigating the negative effects on depressive symptoms in young students. We also propose that there is a direct and inverse effect on depressive symptomatology [38,40,41,42].

## 2. Materials and Methods

### 2.1. Design Research

A non-experimental, predictive and cross-sectional research design was used in this study to examine the effect of emotional exhaustion and optimism on depression in a sample of young people in a trajectory model [43]. The sample was composed of 497 students from a university of middle socioeconomic status from Querétaro city in Mexico. A non-probabilistic sampling strategy was used for selection to generate a sample containing only undergraduate students.

### 2.2. Measurement and Scales 

#### 2.2.1. Patient Health Questionnaire for Depression (PHQ-9)

The PHQ-9 is a screening scale that measures the presence and severity of depressive symptoms [44] and specifically assesses the presence of 9 symptoms in the prior 2 weeks [45]. The scale can be self-administered with scores (not at all = 0, several days = 1, more than half of the days = 2, and almost every day = 3) ranging from 0 to 27 and cut-off points (CP) of 5, 10, 15 and 20, representing levels of depressive symptoms as mild, moderate, moderately severe and severe [42]. These scores can also be used dichotomously for the assessment of depression with a CP threshold of greater than 10 points [41]. The psychometric characteristics of the PHQ-9 indicate Cronbach’s α internal consistency of 0.86–0.89 and construct validity. For our study, we used the Spanish version [42].

#### 2.2.2. Emotional Exhaustion Scale (EES)

This scale is used to measure emotional exhaustion or burnout as an initial response to stress. It contains 10 statements about emotional exhaustion. The answers are based on a Likert scale with evaluation from never = 1 to always = 5, considering the last 12 months of student activity. The scale has concurrent validity with anxiety and an internal structure with values of CFI = 0.90; GFI = 0.89; NFI = 0.88; RMSEA = 0.11. The authors report that Cronbach’s Alpha coefficient value = 0.90. The scale offers values ranging from 10 to 50 points. A score of 26 indicates moderate emotional exhaustion, and a score of 42 indicates high emotional exhaustion [46,47].

#### 2.2.3. Personal Stress Resources Scale

This is measured using a Likert scale and includes two factors: stress and coping [48]. For the first factor, stress, there are six items on feeling overwhelmed and pressure. For the second factor, there are items about whether the person can integrate management stress techniques in their daily life, such as exercise and alternative actions (e.g., talking about their problems). For this study, the Cronbach’s Alpha coefficient value was 0.86 (CI = 0.84–0.88) for stress and was 0.78 (CI = 0.75–0.81) for coping.

#### 2.2.4. Mexican Optimism Scale (MOS)

This scale assesses levels of optimism [3,46]. The MOS consists of 12 items and is rated on a 5-point Likert scale (1 = never, 5 = almost always). In the current study, internal consistency reliability (α = 0.88 [95% CI = 0.87–0.89]) was adequate. Additionally, the CFA indicates adequate construct validity (*CFI* = 0.96, *TLI* = 0.96, *NFI* = 0.95, *IFI* = 0.96, *RMSEA* = 0.05). 

### 2.3. Procedure 

The data was obtained in two months. The instruments were administered to the participants digitally through Google Forms, and the process was carried out in classrooms, through a link and with a QR code to respond, with a response time of 20 min. The form detailed the study’s aim, and participants were asked to provide truthful answers for research purposes.

### 2.4. Statistical Methods 

Descriptive statistics were used for depression, emotional exhaustion, stress, coping and optimism. The Shapiro–Wilk test was used to test for normal distribution, and then non-parametric tests were used. We used Mann–Whitney U and Kruskal–Wallis comparison measures to analyze the differences between demographic variables. Spearman’s Rho correlation was used to assess the relationship between depressive symptoms, fatigue, stress, coping and optimism. Finally, we used path analysis to evaluate the effects of fatigue, stress, coping strategies and optimism resources on depression, as well as to test the moderation of the optimism between coping strategies and depression. We used ML estimation with normal theory for the analysis of moderated models. Mediation analysis was used to determine whether optimism moderated the indirect relationship between stress, fatigue and coping strategies on depressive symptoms [49,50]. Moderated mediation models were interpreted using standardized path estimates (*β*) and squared multiple correlations (*R*^2^). Throughout all analyses, *p* ≤ 0.05 was interpreted as statistically significant. Effect sizes were calculated using conventional metrics [45]. Statistical analyses for the path model were conducted using JASP Program version 0.9.2.

## 3. Results

Table 1 details the sample characteristics, and the participants ranged in age from 17 to 29, with a mean age of 21.32 ± 2.0. Most were women, reported being single and were enrolled in health sciences.

### 3.1. Descriptive Statistics

The descriptive analyses are shown in Table 2. When analyzed, the levels of depression are below the cut-off point = 10 established in the PHQ. However, 28% have a moderate level, 11% are severe and 4.4% are extremely severe. It can also be observed that adaptive coping mechanisms present a higher average compared to stress levels. Specifically, the analysis shows that emotional exhaustion has an above average score, exceeding the theoretical mean. The scores for optimism, on the other hand, are as expected, i.e., close to the theoretical average. The variables present skewness and kurtosis values within −2/+2. Moreover, the sample is larger than 50 (*n* = 497), which ensures statistical robustness for multivariate normal distribution. 

As a secondary analysis, we examined the effect of demographic variables on the study variables. No relationship was found between age and emotional exhaustion (r = −0.08; *p* = 0.06), stress (r = −0.06; *p* = 0.18), coping (r = 0.04; *p* = 0.29) or optimism (r = 0.06, *p* = 0.16), with only a low and negative relationship found with depression (r = −0.10; *p* = 0.02). When analyzing the variables with marital status, no differences were found in terms of depression (H = 3.89; *p* = 0.14), emotional exhaustion (H = 4.63; *p* = 0.09), stress (H = 5.05; *p* = 0.08), coping (H = 4.23; *p* = 0.12) or optimism (H = 3.04, *p* = 0.21). Finally, when comparing men and women, differences were identified only in emotional exhaustion (U = 20,614.000; Z = −3.85; *p* = 0.000; r_b_ = 0.21) and stress (U = 21,859.000; Z = −4.42; *p* = 0.000; r_b_ ^=^ 0.23), with women scoring higher. No differences were found in terms of depression (U = 27,141.500; Z = −0.989; *p* = 0.32), coping (U = 27,962.500; Z = −0.457; *p* = 0.64) or optimism (U = 27,772.000; Z = −4.42; *p* = 0.54).

### 3.2. Correlation Analysis

Table 3 shows the Spearman correlations, which measure the links between depressive symptoms and various fatigue consequences. For example, depression has a strong positive and significant correlation with burnout and stress, as well as an inverse but significant correlation with coping and optimism. Furthermore, it was found that burnout has a high and significant correlation with stress, a moderate and significant correlation with optimism, and a low and negative correlation with coping. Finally, stress was moderate and significant with optimism and was not related to coping. 

### 3.3. Path Analysis 

Table 4 shows the empirically estimated standardized results from the model. The analysis shows that emotional exhaustion and stress significantly predict depression with a direct effect. In addition, the coping strategies used by young people did not have a direct effect on depression (*β* = −0.039; 95% CI = −0.13 to 0.05, *p* = 0.41) in this study.

The indirect effect was tested, and the results suggest that the relationship of stress and coping with depression is mediated by optimism (Table 5). The emotional exhaustion path was not significant (*β* = 0.014; 95% CI = −0.00 to 0.03, *p* = 0.147) in the total model nor for stress in women (*β* = 0.022; 95% CI = −0.15 to 0.058, *p* = 0.24). These factors explained 28.8% of the variance in the optimism resources in the total model, with 30.7% for women and 27.7% for men. 

Table 6 details that the total effects or paths of emotional exhaustion and stress increase depression symptoms in the total model and in women but not in men, for whom lower coping influences the symptoms of depression (*β* = −0.211; 95% CI = − 0.34 to −0.07, *p* = 0.003). Coping decreased and remained significant in controlling optimism. The results of this analysis suggest that the effect of the relationship between emotional exhaustion and stress on depression is partially mediated by optimism in young people. The estimated factors explained 45.0% of the variance in depressive symptoms in the total model (Figure 1), with 45.8% for women and 44.8% for men. 

## 4. Discussion

The objective of this study was to examine the paths of emotional burnout, stress and coping on depressive symptoms, as well as investigate whether optimism would buffer depressive symptoms. The results of this research provide valuable insights into the relationships between emotional burnout, stress, coping, optimism and depressive symptoms in young adults. In terms of the total scores for depressive symptoms, we can highlight that the levels of depression identified in the students of this sample are between moderate (28%) and severe (11%).

When analyzing variables, we found that perceived stress is highly correlated with emotional exhaustion and depressive symptoms. However, no relationship was found with coping. These results can be explained based on previous findings [8,10,51], since for individuals in situations perceived as stressful, the level of stress can become chronic or cause burnout [52]. Additionally, when young people use coping strategies [53], there is an increase in problem-solving alternatives [54], which leads to a decrease in emotional exhaustion and, consequently, better management of negative emotional states or depressive symptoms [5,30,38,41], as shown by the correlation analysis.

### 4.1. Effects in Path Model on Depression 

According to the results of the empirically estimated path model, when analyzing the direct effects, both stress and emotional exhaustion were found to be predictors for depressive symptoms [13,16,51]. However, stress management did not influence the emotional states of depression; this can be explained by the coping strategies evaluated by the instrument being aimed at managing stress, such as through doing physical activity, taking it easy or talking about problems [5,30,55,56]. However, within the contextual model of depression, these same behaviors can be considered avoidant behaviors that maintain depressogenic behaviors, since they are not modifying the precipitating situations that elicit the behaviors linked to depressive emotional states. Therefore, as the behavior is not aimed at establishing motivating objectives or goals for the student, the performance of said avoidant behaviors will continue [21,22]. 

Regarding indirect effects, the coefficient analysis indicates that optimism mediates the relationship between stress, coping and depressive symptoms (Table 4). This suggests that optimism can mitigate the negative impact of stress and ineffective coping strategies on mental health [22,31]. These results may reflect that individuals with higher levels of optimism are better equipped to handle stress and employ adaptive coping mechanisms, thereby reducing their risk of developing depressive symptoms [56,57,58]. Consistent with previous studies [3,38,40,41,42,46], these results [40] support the idea that imagining successful and positive situations reinforces the vision of an optimistic and better future. For example, a young person who behaves with enthusiasm when facing complicated situations, or who is cheerful when faced with difficult situations, is very likely to maintain hope when facing difficult moments they may encounter in their life, and the above would explain the onset or prognosis of depressive symptoms in young people. Therefore, partial evidence is found to support our hypothesis that optimism factors are mediators and contribute to the reduction in depression. Contrary to expectations, optimism was not statistically significant as a mediator in the direct relationship between emotional exhaustion and depressive symptoms. This could be due to several factors, such as the specific measures used, the sample characteristics or the complex interplay between burnout and other variables. It is important to note that while emotional exhaustion may have a direct impact on depression in this study, it could still indirectly influence mental health through other factors, such as increased stress or impaired coping abilities. Young people with emotional exhaustion are likely to report feeling emotionally drained, to have days when they do not sleep well because of studying, and to experience a low mood or feel sad for no apparent reason [17,20,46,47], and this would explain the strong association with depressive symptomatology, where a positive outlook from optimism [3,46] is not enough to cope with life circumstances. 

In contrast to the indirect effects, the findings revealed that optimism partially mediates the relationship between emotional exhaustion and stress on the one hand, and depressive symptoms on the other hand. This result is significant, as it provides empirical evidence for the mechanism through which optimism influences the vulnerability to manifesting symptoms of depression in this age group [32,34,38,41,58]. Specifically, our results showed that although emotional exhaustion and stress exert a direct and significant influence on depression, this effect is diminished by the presence of optimism. The fact that coping capacity decreased but remained significant even when affected by optimism suggests that the latter does not explain the entire relationship between stress/exhaustion and depression but rather operates as a partial mediator [22,30,31,32,33,35]. Regarding the partial mediation effect of optimism, we propose that it can be interpreted as follows: Young people with higher levels of optimism tend to cope with stressful situations and experiences of emotional exhaustion in an adaptive way. They are likely to resort to coping strategies such as seeking social support, problem solving and positive reinterpretation of situations, which in turn reduces the likelihood of developing depressive symptoms [5,30,38,41]. In contrast, youth with low levels of optimism may be more likely to experience hopelessness, rumination and avoidance, which increases the negative impact of stress and emotional exhaustion on their personal well-being. A second explanation is the implication that there are other pathways, besides optimism, through which stress and emotional exhaustion impact the mental health of young people [56,57,58]. However, in both scenarios, the observed partial mediation underlines the importance of optimism as a relevant protective factor [3,46]. Finally, in this study, 45% of the depressive symptoms are explained by the set of factors estimated in our model, which is why we consider it a very significant finding for explaining depressive symptomatology. The proposed model indicates that the incorporation of emotional exhaustion, stress and optimism as mediators provides considerable insight into the factors contributing to depression in young people. Although the percentage of variance remains unexplained, this result suggests that the model captures a significant proportion of the complexity of this phenomenon. 

### 4.2. Limitations and Suggestions 

The study has some limitations, for example, that its cross-sectional design prevents us from demonstrating a clear cause-and-effect connection, although the trajectory model points to the effect of the independent variables (stress, burnout and optimism) on the dependent variables (depressive symptomatology). Future research incorporating a longitudinal design will be necessary to confirm the long-term effects and the direction of the relationships observed in our study. In addition, quasi-experimental studies could be conducted, aimed at promoting optimism to reduce depressive symptomatology in young people. A second limitation is the size (*n* = 497) and representativeness of the sample. In terms of size, although it is a relatively large sample, it cannot be considered representative of the population of young Mexicans, because the participants were young people non-randomly selected from the central lowlands of Mexico, rather than from various cities in the country. This reduces the sample representativeness, so it is necessary to interpret the results with caution. The last limitation that we consider relevant is the measurement scale used for stress and coping [48]. Although it is a scale that has been used in previous studies [55,56] with satisfactory results, it does not incorporate the physiological stress responses that could manifest in young people when facing the different daily stressors they are exposed to at this stage of life. Regarding coping, the scale incorporates some social support coping strategies (e.g., talking about how I feel to someone I trust to eliminate stress) but more avoidance strategies, as used in other measurement scales [57,59]. Hence, in future studies, we recommend using different measurement scales that more accurately assess the different coping strategies to corroborate our results. 

The study provides valuable evidence for the role of optimism as a mediator in the relationship between emotional exhaustion, stress and depressive symptoms in young people. The results underline the importance of promoting optimism as a key psychological resource to improve the mental health of this age group. To further explore these findings, future research could study the underlying mechanisms through which optimism buffers the relationship between stress, emotional burnout and depression in addition to incorporating other variables, such as gratitude [49,58] or kindness [50,58,59], that may be contributing via having a moderating effect in reducing depressive symptoms in young people. By addressing these research questions, we can gain a deeper understanding of the factors that contribute to mental health problems among young adults and develop more effective interventions to promote well-being. 

### 4.3. Practical Implications 

The results obtained in this study have important implications for young people regarding their development into adulthood. Identifying such factors as optimism, which modulate the transition to emotional states of depression from situations perceived as stressful, is helpful for preventive approaches in a problem-solving model. Optimism, being related to stable and generalized expectations that positive things will happen in the future, can be reflected in having a positive orientation to problems, which represents the beginning of the problem-solving process for the development of appropriate coping strategies [3,46]. In contrast, for chronic levels of stress, this modulating impact is not observed in the transition to depression, since some clinical mental health problems may have developed that require a therapeutic approach derived from the time in which the individual was subjected to these chronic levels of stress. Therefore, there is development of a negative orientation to the problem and a deficit of coping strategies such as problem solving.

The results demonstrate the importance of effective coping strategies based on optimism, but also the need to develop intervention programs that teach young people skills to manage stress and adversity, focused on acting towards modifying their environment to reduce the impact on their unpleasant emotional states and therefore on problem-solving patterns.

Ultimately, our findings contribute to a better understanding of the factors that influence the mental health of university students within the Latin American cultural context and provide a basis for designing preventive interventions for the future. Interventions aimed at promoting optimism in young people could be an effective strategy to prevent and reduce depressive symptoms, especially in those who experience high levels of stress and emotional exhaustion. These could include training programs for positive psychology and techniques that facilitate coping skills in young people. Optimism training in youth is a promising strategy for strengthening emotional well-being. We propose three practical ways to increase optimism. First, young people could engage in regular gratitude exercises [60] to cultivate appreciation for the positive aspects of their lives. A second way is through promoting their strengths, and we propose demonstrating to young people how to focus on their own strengths and personal resources (optimism) to face their challenges [5]. The young people would be taught individually or in groups to identify the skills they have or have used in the past to overcome obstacles similar to those they may face [61]. Finally, young people can be instructed to practice visualizing not only the desired outcome but also the process and using their own strengths to achieve it, as demonstrated in previous studies [62,63]. For our research team, we have set the short-term objective of conducting studies or interventions to track changes in these variables over time and to assess their long-term impact.

## 5. Conclusions

The results demonstrate that emotional exhaustion and stress significantly and directly predict depression. However, coping strategies did not have a direct effect on depression in this sample. Optimism significantly mediated the relationship between stress and depression. The mediating role of optimism was not as significant in the pathway involving emotional exhaustion as it was in that involving coping. This may imply that higher levels of optimism are needed to mitigate the negative impact of stress and the indirect effect of coping with depressive symptoms in young university students.

This finding highlights the complex interplay of these factors and suggests that the impact of emotional exhaustion on depression might operate through different mechanisms than those of stress.

These findings underscore the importance of fostering optimism in young university students. Interventions that promote optimism and more effective coping mechanisms may be successful in reducing the risk of depression, particularly in the context of academic stress and emotional exhaustion. Further research is needed to understand the specific mechanisms through which optimism exerts its protective or buffering effects and to explore the implications of these findings as a basis for the development of targeted interventions for young people.

## Figures and Tables

**Figure 1 brainsci-15-00656-f001:**
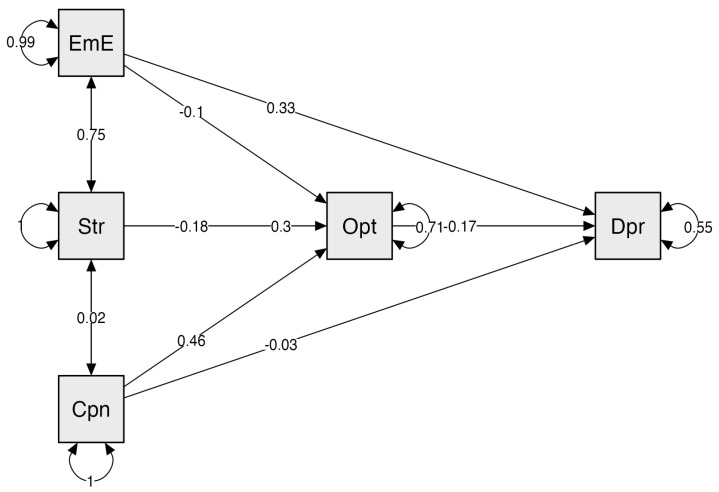
Path model effects of emotional exhaustion, stress, coping and optimism on depression. Note: The path model shows that optimism mediates the effect of stress and coping and partially mediates the relationship with emotional exhaustion and its effect on depressive symptoms. EmE = emotional exhaustion. Str = stress. Cpn = coping. Opt = optimism. Dpr = depression.

**Table 1 brainsci-15-00656-t001:** Sample characteristics.

	*Frequency*	*Percentage*
*Sex/gender*		
Men	182	36.6
Women	315	63.4
*Marital status*		
Single	453	91.1
Married	13	2.6
Cohabitating	30	6.0
*Field of study*		
Health sciences	231	46.5
Social sciences	40	7.9
Arts and humanities	59	11.9
Business	65	13.0
Tourism and gastronomy	9	1.8
Engineering	73	14.7
Communication and multimedia	20	4.2

**Table 2 brainsci-15-00656-t002:** Descriptive analysis of the variables being studied.

	*M*	*SD*	*Min*	*Max*	*S*	*K*	*α Reliability*
Depression	8.77	5.7	0.00	26.00	0.439	−0.236	0.87
Stress	17.48	4.8	6.00	30.00	0.122	−0.144	0.86
Coping	19.78	4.2	6.00	30.00	0.002	0.283	0.78
Emotional exhaustion	30.08	7.4	10.00	50.00	0.159	0.021	0.88
Optimism	30.75	6.8	9.00	45.00	0.088	−0.197	0.91

Note: *M* = Mean; *SD* = Standard Deviation; *Min* = Minimum; *Max* = Maximum; *S* = Asymmetry; *K* = Kurtosis.

**Table 3 brainsci-15-00656-t003:** Spearman’s correlations between the study variables.

	Emotional Exhaustion	Stress	Coping	Optimism
Depression	0.604 ***	0.576 ***	−0.223 ***	−0.400 ***
Emotional Exhaustion		0.748 ***	−0.128 ***	−0.290 ***
Stress			−0.046	−0.265 ***
Coping				0.507 ***

Note: *** *p* < 0.001.

**Table 4 brainsci-15-00656-t004:** Direct effects on depression.

						Confidence Intervals 95%	
Paths			*β*	SE	Z	Lower	Upper
Emotional Exhaustion	→	Depression	0.335	0.063	5.29 ***	0.211	0.458
Stress	→	Depression	0.357	0.065	5.47 ***	0.229	0.485
Coping	→	Depression	−0.031	0.035	−0.89	−0.100	0.037

Note: *** *p* < 0.001.

**Table 5 brainsci-15-00656-t005:** Indirect effects of optimism on depression.

								Confidence Intervals 95%	
Paths					*β*	SE	Z	Lower	Upper
Emotional Exhaustion	→	Optimism	→	Depression	0.018	0.012	1.45	0.006	0.042
Stress	→	Optimism	→	Depression	0.032	0.014	2.29 *	0.005	0.059
Coping	→	Optimism	→	Depression	−0.079	0.020	−3.98 ***	−0.118	−0.040

Note: * *p* < 0.01, *** *p* < 0.001.

**Table 6 brainsci-15-00656-t006:** Total effects of emotional exhaustion, stress and coping on depression.

						Confidence Intervals 95%	
Paths			*β*	SE	Z	Lower	Upper
Emotional Exhaustion	→	Depression	0.353	0.064	5.52 ***	0.227	0.478
Stress	→	Depression	0.328	0.056	5.88 ***	0.219	0.438
Coping	→	Depression	−0.111	0.032	−3.46	−0.173	−0.048

Note: *** *p* < 0.001.

## Data Availability

The datasets generated during the investigation are available from the corresponding author. The data are not publicly available due to ethical reasons.

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
