# Peer review of "Trajectories of Emotional Exhaustion and Their Contribution to Depression: Optimism as a Buffer in Young People"

_brainsci, 2025, doi:10.3390/brainsci15060656_

Round 1
Reviewer 1 Report
Comments and Suggestions for Authors
Dear authors,
It has been a pleasure for me to review this work that deals with optimism as a buffer against depression in young people. I found it very interesting.
With the sole purpose of improving the quality of this manuscript, I will allow myself to make some comments:
- In the introduction, section 1.4. Depressive symptomatology, depressive symptomatology should be described according to the corresponding manuals, such as the DSM-5 (Diagnostic and Statistical Manual of Mental Disorders, Fifth Edition).
- The last section of the introduction should be placed at the end of the discussion.
- Section 2. Materials and Methods. It is necessary to define the type of study, how the non-probability sample was recruited, and clarify where (it is stated that it was at one university, while another section states that it was at two).
- Section 2.2.1 Patient Health Questionnaire-9 for depression (PHQ-9). This should be further developed to understand the questionnaire.
- Section 2.2.2 Emotional Exhaustion Scale (EES). This should be further developed.
- The description of the sample (age, gender, etc.) does not belong in this section but rather in the results section and should be explained in a table.
- Table 1: Descriptive analysis of the variables being studied. It's not descriptive, it's numerical. A truly descriptive table would have to be created so we could read it and understand how it behaves, and what each variable is like.
- On page 6: "The results of this analysis suggest that the relationship between stress and coping with depression is mediated by optimism (Table 4)." Explain how they are related.
- On page 10: "The results underline the importance of promoting optimism as a key psychological resource to improve the mental health of this age group." Explain how the promotion of optimism is proposed.
- The conclusions should be written according to the established objectives of the study.
All the best.
Author Response
Dear reviewer, we appreciate your comments, which helped us improve the document. Below are the responses to your review.
- In the introduction, section 1.4. Depressive symptomatology, depressive symptomatology should be described according to the corresponding manuals, such as the DSM-5 (Diagnostic and Statistical Manual of Mental Disorders, Fifth Edition).
Response: The symptomatology is described with the DSM - 5 as proposed
- The last section of the introduction should be placed at the end of the discussion.
Response: This section was modified from the introduction to the discussion
3, Section 2. Materials and Methods. It is necessary to define the type of study, how the non-probability sample was recruited, and clarify where (it is stated that it was at one university, while another section states that it was at two).
Response: the type of study and sample selection were incorporated
4, Section 2.2.1 Patient Health Questionnaire-9 for depression (PHQ-9). This should be further developed to understand the questionnaire.
Response: The explanation of the instrument was better developed.
- Section 2.2.2 Emotional Exhaustion Scale (EES). This should be further developed.
Response:: The explanation of the instrument was better developed.
- The description of the sample (age, gender, etc.) does not belong in this section but rather in the results section and should be explained in a table.
Response::The sample description is changed to the results section as directed by the reviewer.
- Table 1: Descriptive analysis of the variables being studied. It's not descriptive, it's numerical. A truly descriptive table would have to be created so we could read it and understand how it behaves, and what each variable is like.
Response: A descriptive analysis is considered to be a statistic that incorporates measures of central tendency, dispersion and normality of the variables.
- On page 6: "The results of this analysis suggest that the relationship between stress and coping with depression is mediated by optimism (Table 4)." Explain how they are related.
Response: The explanation is in the discussion, sección 4.2. Indirect effects optimism on depression
- On page 10: "The results underline the importance of promoting optimism as a key psychological resource to improve the mental health of this age group." Explain how the promotion of optimism is proposed.
Response:: Ideas for how to do this are described in the discussion in section 4.5. Practical Implications, and three examples for implementation are included.
- The conclusions should be written according to the established objectives of the study.
Response:: The discussion was adjusted according to the objectives of the study.

Reviewer 2 Report
Comments and Suggestions for Authors
- “There is a high prevalence of emotional exhaustion among students as they progress in their curricula (8)(6)(9) and deal with due dates for assignments and exams as they also bear with lifestyle changes, peer pressure, and financial struggles (7)(10).” Could you please provide the exact report prevalence or a range of prevalence? Same to “Previous studies have documented an increase in depression rates (28)(30) among university students…”.
- Overall, I understand that the introduction had to cover all four parts and their connections. However, its length—particularly due to information not directly tied to the main research question—may make it harder for readers to stay focused. I recommend streamlining the introduction to focus more directly on the relationships between stress, emotional exhaustion, depression, and optimism, and suggest that background details (e.g., lines #72–81) or some content that may be duplicated (e.g., lines 108-110) could be shortened or rephrased for clarity and conciseness.
- “By identifying the mediating role of optimism 115 on depressive symptoms …” The sentence feels incomplete. As mediator should be introduced with both a predictor and an outcome, I suggest explicitly stating both variables involved.
- “The sample was composed of 497 students recruited at university in the Querétaro 123 city in México.” Were these undergraduate or graduate students? If all participants belong to the same educational level (e.g., all undergraduates), please clearly state this in the Methods section. If the sample includes students from different educational levels, what were the proportions of undergraduates, master’s, and PhD students? I expect stress levels may differ across these groups, so I recommend reporting stress and depression, etc., outcomes by educational level in the Results section. Also, if they are from different educational levels, consider the potential impact of educational level on the association between stress and depression, as well as the mediation of optimism.
- The authors reported that participants ranged in age from 18 to 29, with a gender imbalance and differences in marital or living status. However, they did not examine the potential associations between gender, age, or living status and the key variables—depression, stress, coping, emotional exhaustion, and optimism. Based on the Methods and Results sections, I found no evidence that the authors accounted for the possible influence of these demographic factors on either the associations among the key variables or the mediating role of optimism. For example, being single versus partnered could potentially moderate how optimism influences the relationship between stress, coping, emotional exhaustion, and depression.
- Table2, Given the number of tests, I’d suggest applying FDR or other multiple comparison corrections to them.
- The path model doesn’t look very clear to me. Please add a figure legend to explain it.
- Discussion: The discussion provides interpretation of the findings, but it is somewhat repetitive, as similar points are restated across the direct, indirect, and total effects subsections. For example, the mediating role of optimism is described in both the “Indirect Effects” (lines 268–276) and “Total Effects” (lines 296–315) sections with mostly overlapping wording. While I understand the intention to structure the discussion around direct, indirect, and total effects, the overlapping interpretations could be streamlined. I suggest reducing redundancy within and across these sections while keeping the structure to enhance clarity and avoid unnecessary repetition.
Author Response
- “There is a high prevalence of emotional exhaustion among students as they progress in their curricula (8)(6)(9) and deal with due dates for assignments and exams as they also bear with lifestyle changes, peer pressure, and financial struggles (7)(10).” Could you please provide the exact report prevalence or a range of prevalence? Same to “Previous studies have documented an increase in depression rates (28)(30) among university students…”.
Response: the requested information was incorporated
2. Overall, I understand that the introduction had to cover all four parts and their connections. However, its length—particularly due to information not directly tied to the main research question—may make it harder for readers to stay focused. I recommend streamlining the introduction to focus more directly on the relationships between stress, emotional exhaustion, depression, and optimism, and suggest that background details (e.g., lines #72–81) or some content that may be duplicated (e.g., lines 108-110) could be shortened or rephrased for clarity and conciseness.
Response: Irrelevant information was reduced and redundant information was avoided in the introduction.
- “By identifying the mediating role of optimism 115 on depressive symptoms …” The sentence feels incomplete. As mediator should be introduced with both a predictor and an outcome, I suggest explicitly stating both variables involved.
Response: a better explanation was made of the mediating effect of optimism on depression
- “The sample was composed of 497 students recruited at university in the Querétaro 123 city in México.” Were these undergraduate or graduate students? If all participants belong to the same educational level (e.g., all undergraduates), please clearly state this in the Methods section. If the sample includes students from different educational levels, what were the proportions of undergraduates, master’s, and PhD students? I expect stress levels may differ across these groups, so I recommend reporting stress and depression, etc., outcomes by educational level in the Results section. Also, if they are from different educational levels, consider the potential impact of educational level on the association between stress and depression, as well as the mediation of optimism.
Response: All students were university students and had the same educational level and was clarified in the method section.
- The authors reported that participants ranged in age from 18 to 29, with a gender imbalance and differences in marital or living status. However, they did not examine the potential associations between gender, age, or living status and the key variables—depression, stress, coping, emotional exhaustion, and optimism. Based on the Methods and Results sections, I found no evidence that the authors accounted for the possible influence of these demographic factors on either the associations among the key variables or the mediating role of optimism. For example, being single versus partnered could potentially moderate how optimism influences the relationship between stress, coping, emotional exhaustion, and depression.
Response: The primary objective of the study does not include analyzing the effect of demographic factors on the study variables. However, demographic factors were analyzed as proposed by the reviewer. In our analysis, no relevant association was found, except for differences between men and women in emotional exhaustion and stress, favoring women. The statistics obtained are shown below:
Differences between sexes: Depression (U = 27141.500; Z = -.989; p = .322); Emotional exhaustion (U = 20614.000; Z = - 3.85; p = .000); Stress (U = 21 859.000; Z = - 4.42; p = .000); Coping (U = 27962.500; Z = -.457; p = .648); Optimism (U = 27772.000; Z = - 4.42; p = .540).
Differences with marital status (ANOVA) : Depression (F = 1.64; p = .193); Emotional exhaustion (F= 2.53; p = .080); Stress (F = 2.22 ; p = .109); Coping (F = 2.63; p = .073); Optimism (F = 1.22, p = .295).
Correlation with age: Depression (r = -.10 p = .02); Emotional exhaustion (r = -.08; p = .06); Stress (r = -.06 ; p = .18); Coping (r = .04; p = .29); Optimism (r = .06, p = .16).
- Table2, Given the number of tests, I’d suggest applying FDR or other multiple comparison corrections to them.
Response: It's not clear to us what type of analysis you suggest applying. We can perform additional comparisons if necessary.
- The path model doesn’t look very clear to me. Please add a figure legend to explain it.
Response: A note was added explaining the trajectory model.
Path model demuestra that optimism modia el efecto del estres y el afrontamiento y partially mediates the relationship between emotional exhaustion on depressive symptoms.
- Discussion: The discussion provides interpretation of the findings, but it is somewhat repetitive, as similar points are restated across the direct, indirect, and total effects subsections. For example, the mediating role of optimism is described in both the “Indirect Effects” (lines 268–276) and “Total Effects” (lines 296–315) sections with mostly overlapping wording. While I understand the intention to structure the discussion around direct, indirect, and total effects, the overlapping interpretations could be streamlined. I suggest reducing redundancy within and across these sections while keeping the structure to enhance clarity and avoid unnecessary repetition.
Response:,We believe that in both sections we provide different explanations of what was found, however, we reviewed the sections and removed some parts to avoid redundancy.

Round 2
Reviewer 1 Report
Comments and Suggestions for Authors
I have nothing more to add to this review.
Author Response
We appreciate your notification and attach the version of the second revision.

Reviewer 2 Report
Comments and Suggestions for Authors
I appreciate the authors’s responses and revisions. Please find my responses below.
1. Reply to the authors’ response #4: “All students were university students and had the same educational level and was clarified in the method section.” “The sample was composed of 497 students from 128 a university of middle socioeconomic status from Querétaro city in México.”
Based on the authors’ reply and the revised text, it seems like all students are undergraduate students? If so, please specify this specifically in the main text, at least in the Method section. The term the authors use currently - “university students” - does not specifically refer to undergraduates — it is a broad term.
2. “The primary objective of the study does not include analyzing the effect of demographic factors on the study variables. However, demographic factors were analyzed as proposed by the reviewer. In our analysis, no relevant association was found, except for differences between men and women in emotional exhaustion and stress, favoring women.”
I was previously concerned about how demographic factors might affect the main psychological outcomes because they could act as confounders and skew the relationships between the main constructs being studied. According to the authors' response, there is a significant correlation between sex and stress and emotional exhaustion. In light of this, it would be worthwhile to investigate whether the suggested path model (Figure 1) is consistently valid for both sexes. I recommend that the authors test the model for stress and emotional exhaustion separately in male and female participants to assess potential differences in pathway strength or structure.
Please include the results of these demographic associations and the sex-specific path analyses, either in the main manuscript or as supplementary material. This addition would strengthen the findings' robustness.
3. In response to authors’ response #6: What I suggested is that 10 tests were applied in the same group of participants (in Table3), so I was a bit concerned about the false positive results. The authors could apply multiple comparison corrections (such as Bonferroni correction and Benjamini-Hochberg) to control for false positive results (Type I errors) when performing multiple statistical tests on the same dataset.
This is technically very straightforward. Most statistical software packages offer built-in functions that can apply corrections with a single line of code. Like p.adjust() in R. Ideally, the authors could show the correlation r and corrected P value in each cell in Table 3. For example: r=0.604, PFDR=XXX, where the PFDR means FDR-corrected P value and XXX is the exact value.
Author Response
- Based on the authors’ reply and the revised text, it seems like all students are undergraduate students? If so, please specify this specifically in the main text, at least in the Method section. The term the authors use currently - “university students” - does not specifically refer to undergraduates — it is a broad term.
Response: The term university is changed to undergraduates, to to avoid confusion with the postgraduate students.
- Please include the results of these demographic associations and the sex-specific path analyses, either in the main manuscript or as supplementary material. This addition would strengthen the findings' robustness.
Response: The primary objective of the research remains the same; we do not seek to understand the effect of demographic variables or a differential effect by sex. The requested analyses were integrated into the article.We incorporated into the text the parameter for men and women that was relevant in the model,
3.The authors could apply multiple comparison corrections
Response: We appreciate your comment about administering multiple scales to the same group of participants and share your concern about possible false positive results. We therefore recommend considering the results with caution, and we have addressed this observation in the discussion.